# GaN Heterostructures as Innovative X-ray Imaging Sensors—Change of Paradigm

**DOI:** 10.3390/mi13020147

**Published:** 2022-01-19

**Authors:** Stefan Thalhammer, Andreas Hörner, Matthias Küß, Stephan Eberle, Florian Pantle, Achim Wixforth, Wolfgang Nagel

**Affiliations:** 1Experimental Physics I, University of Augsburg, Universitätsstrasse 1, D-86159 Augsburg, Germany; andreas.hoerner@physik.uni-augsburg.de (A.H.); matthias.kuess@physik.uni-augsburg.de (M.K.); stephan.eberle@student.uni-augsburg.de (S.E.); achim.wixforth@physik.uni-augsburg.de (A.W.); 2Senray Technologies GbR, Ebenböckstrasse 23, D-81241 Munich, Germany; wwnagel@senray.de; 3Walter Schottky Institut, Technical University Munich, Am Coulombwall 4, D-85748 Garching, Germany; florian.pantle@wsi.tum.de

**Keywords:** GaN-HEMT mesa structures, 2DEG, X-ray sensor, X-ray imaging

## Abstract

Direct conversion of X-ray irradiation using a semiconductor material is an emerging technology in medical and material sciences. Existing technologies face problems, such as sensitivity or resilience. Here, we describe a novel class of X-ray sensors based on GaN thin film and GaN/AlGaN high-electron-mobility transistors (HEMTs), a promising enabling technology in the modern world of GaN devices for high power, high temperature, high frequency, optoelectronic, and military/space applications. The GaN/AlGaN HEMT-based X-ray sensors offer superior performance, as evidenced by higher sensitivity due to intensification of electrons in the two-dimensional electron gas (2DEG), by ionizing radiation. This increase in detector sensitivity, by a factor of 10^4^ compared to GaN thin film, now offers the opportunity to reduce health risks associated with the steady increase in CT scans in today’s medicine, and the associated increase in exposure to harmful ionizing radiation, by introducing GaN/AlGaN sensors into X-ray imaging devices, for the benefit of the patient.

## 1. Introduction

Until recently, the commercialization of gallium nitride (GaN) devices was hindered by economic difficulties in the scale fabrication of GaN crystals. However, with the improvement of GaN fabrication, from 2010 to 2016, to a high-quality material, these difficulties have been overcome by various advances in metal-organic chemical vapour deposition (MOCVD) and molecular beam epitaxy (MBE) processing techniques; GaN-based devices are now even able to significantly outperform their Si-based counterparts. With the improvement of GaN material in form of GaN-wafers, the detection of X-rays by GaN sensors can be further explored, and is now on the threshold of technology transfer to society. Due to the inherent features of two-dimensional electron gas (2DEG) in GaN high-electron mobility transistors (HEMT), the intrinsic enhancement by a factor of 10^4^ for the number of electrons that can be collected from single ionization events by direct conversion photon counting reveals the tremendous potential of GaN-sensors compared to current technologies, which use indirect conversion systems. These breakthrough findings paved the way for a paradigm shift in X-ray detector technology, while enabling a breakthrough in X-ray and computer tomography (CT) examinations by reducing the effective dose, used today in routine X-ray and CT examinations, to 1/20.

## 2. GaN Manufacture and *GaNification*

Polycrystalline GaN material was synthesized for the first time around 1930 by flowing ammonia (NH_3_) over liquid gallium (Ga), at around 1000 °C [1]. The first single crystalline GaN was epitaxially grown in 1969 by Maruska and Tietjen [2] by hydride vapor phase epitaxy (HVPE) on a sapphire substrate [3]. In this process, gaseous hydrogen chloride initially reacts with liquid Ga at a temperature of approximately 880 °C to form gallium chloride (GaCl_3_). In a reaction zone, the GaCl_3_ is brought close to a GaN crystal nucleus at temperatures between 1000 and 1100 °C. Here, the GaCl_3_ reacts with the inflowing ammonia, releasing HCl to form crystalline GaN, which crystallizes preferably in the (hexagonal) wurtzite structure [4].

GaN crystals can be grown on a variety of substrates, including sapphire, silicon carbide (SiC) and silicon (Si). By growing a GaN epi layer on top of silicon, the existing silicon manufacturing infrastructure can be used, eliminating the need for costly specialized production sites, and leveraging readily available large-diameter silicon wafers at low cost.

In 1993, using metal organic chemical vapor epitaxy (MOVPE), Asif Khan et al. [5] grew the first AlGaN/GaN heterojunction. Despite a moderate crystallographic quality, the mobility of the 2DEG, at the AlGaN/GaN interface was around 600 cm^2^/Vs. This achievement can be regarded as the start of a quest for nitride high-electron mobility transistor technology [6].

In 1999, Ambacher and colleagues [7] proposed a model to analytically describe the 2DEG properties of AlGaN/GaN heterostructures. Today, this model has been widely adopted by the GaN community. In the same year, Sheppard et al. [8] demonstrated high-power microwave HEMTs based on AlGaN/GaN heterostructures grown on a silicon carbide (SiC) substrate.

In 2000, the nature of the 2DEG was further clarified by Ibbetson et al. [9], attributing a key role to the surface states present in nitride materials as source of electrons. A listed historical summary can be found by Roccaforte and Leszczynski [10]. Binary AlN/GaN HEMTs are preferably grown by plasma-assisted molecular beam epitaxy (PA-MBE), as well as metal-organic chemical vapour deposition (MOCVD), which is also known as metal-organic chemical vapour phase epitaxy (MOCVPE) [11,12]. The latter method, which is likely to dominate by more than one magnitude, has gained increased attraction, since MOCVD offers simple fabrication of large quantities, as requested in commercial use for the production of complex semiconductor structures [13] (Figure 1).

In contrast to the well-known silicon, the structure of these HEMT comprise not only from a single element, but from at least two, or more, including gallium arsenide (GaAs), gallium nitride (GaN), indium phosphide (InP) and aluminium nitride (AlN). Since they are arranged according to the periodic table, from rows III. and V. of the main group, they are called III-V semiconductors. A 2DEG is inherently present in the AlGaN/GaN heterostructure, the presence of which is the basis for the operating principle of HEMT devices [10]. HEMT devices were structured from 3μm thick Ga-face GaN films that were grown by MOCVD (TopGaN Ltd., Warsaw, Poland) on a 330 μm thick c-plane sapphire substrate, followed by a 2.8 µm GaN layer, a 25 nm Al_0.25_Ga_0.75_N layer, a 3nm nominally undoped GaN capping layer, and an 88 nm isolation layer. The 2DEG formed just below the heterojunction interface had a sheet carrier concentration of n_2DEG_ ≈ 8 Å~1012 cm^−2^ and a mobility of μ ≈ 1100 cm^2^/Vs at 300 K. In-depth fabrication methods and physical properties of the HEMT devices are described elsewhere [14,15]. Photolithography and reactive ion etching steps were performed according to standard procedure described in detail in Howgate et al. [16]. The ohmic contacts with 200/800/100/900 Å stacks of Ti/Al/Ti/Au defined by lithography were fabricated by evaporation steps [15] (Figure 1). This process produces contact electrodes without affecting the 2DEG conduction.

One of the key features of these heterostructures is the formation of the 2DEG and the resulting properties. The simulation of the Al content in the Al-rich heterostructure (software “Nextnano”) showed dependence on the Al concentration—the lower the Al content in the channel, the higher the electron density—and reached significant values well-above 2 × 10^13^/cm^3^ [17]. The band gap of the AlGaN layer can be varied by changing the ration of Al to Ga present within the AlGaN layer. The bandgap varies approximately in accordance with Vengard’s Law [18]. A striking feature of the lateral GaN HEMTs is the zero-charge feedback. Due to the absence of p-n junctions and current flow in a polarization induced 2DEG, reverse operation starts when the drain voltage falls below the sum of the gate potential and the threshold voltage, thus creating a reverse channel. Furthermore, due to the expansion of the space charge layer along GaN/AlGaN interface within a basically undoped substrate, the output capacitance shows a very linear characteristic [19].

Since the 1990s, GaN has been used in opto-electronics, commonly in light emitting diodes (LED). GaN emits a blue light, used for disc-reading in Blu-ray. Additionally, GaN is used in semiconductor power devices, RF components, lasers, and photonics with an overall reduction of electric power consumption. However, superior features of GaN-based HEMTs compared to Si-based devices, such as faster switching speed, higher thermal conductivity, and lower on-resistance, outcompete Si-based solutions in many areas. Hence, the overall nitride device market forecasts for the next years are much brighter compared to those of the other compound semiconductors. According to the market analysis report of Grand View Research, the global GaN semiconductor devices market size was valued at USD 1.65 billion in 2020 and is expected to expand at a compound annual growth rate (CAGR) of 21.5% from 2021 to 2028 [20]. Their wide bandgap and high stability make them a perfect material for high-power and high frequency transistors present in the market. For those reasons, the term “*GaNification*” was created for the revolution expected in modern electronics and opto-electronics [10].

In our view, and by analogy, a change of paradigm can be expected from GaN-based X-ray detector technology for the development of GaN-based devices described above. In contrast to the notion that GaN layers are poorly suited for X-ray dosimetry, held back in 2008, it can be demonstrated that a GaN thin film arranged on a carrier substrate with a thickness of <50 µm, provided with ohmic contacts, allows a sensitive, reproducible direct detection of X-rays with an energy of 1-300 keV (up to values in the µGy range) [21,22]. The functional principle of the GaN thin film radiation detector fundamentally differs from the conventionally available direct semiconductor detectors for X-ray radiation, as it requires a simple resistance or conductivity measurement, such as a photoconductor [23].

With regard to the application of X-rays in today’s medicine, CT scanning is the method of choice. CT scanners use computer-controlled X-rays to create images of the body. At the same time, the effects of radiation and increasing health concerns hinder the market growth of CT scanner devices and equipment. Radiation coming from manmade sources, such as CT scans, nuclear medicine scans, and PET scans, carry major health hazards and risks. Accumulated low doses can cause cancer in the long run. The medical justification for performing CT imaging is that the anticipated benefit exceeds the anticipated radiation exposure risk. Thus, an optimal dose of radiation to accomplish the diagnostic task should be used. However, this optimal dose is not always known to the practitioner, resulting in a wide spectrum of radiation doses being used, even for common diagnostic tasks [24,25].

The increasing awareness of risks associated with radiation exposure triggered a variety of dose-reduction techniques, such as tube current or tube voltage modulation, automatic exposure control, and low-kilovolt scanning [26], where a dose reduction between 10% and 30% is possible, depending on examination type. Compared to the real-time analytical reconstruction method, filtered back projection (FBP) [27], which has reached its limitation and does not allow for further dose reductions, vendor-specific iterative reconstruction algorithms were introduced, which allow further dose reductions [28,29].

Altogether, there remains a strong medical need for means to reduce the effective dose in CT-scanners. We address this need with a proprietary detector technology based on GaN thin film. Compared to current CT detectors that use indirect converter systems, direct converter photon counting results in a breakthrough for X-ray and CT-scans in reducing the photon energy, from up to 130 keV [30] to 5 keV [16]. The effective dose can be minimized in the state-of the art dose applied in the routine X-ray and CT-scans of today. In the following section we describe the working principle and imaging potential of GaN-HEMT mesa structures.

## 3. X-ray Detection Using Semiconductor Material

In 1946, R.S. Ohl discovered the p-n junction in silicon, which can be used for particle detection [31]. Electron–hole pairs are the basic information carriers in semiconductor detectors. Electromagnetic radiation (X-ray and gamma ray) and particle radiation (alpha and beta radiation) generate free charge carriers, electrons, and holes in the semiconductor along the pathway taken through the detector. By capturing these electron–hole pairs, the detection signal is formed (Figure 2). Whether, and how many, electron–hole pairs are generated by the incident ionizing radiation depends largely on the size of the band gap energy. This band gap, or forbidden zone, describes the energetic distance between the valence and conduction band. When photonic energy is supplied to the band gap area, valence electrons are excited into the conduction band. These electrons in the conduction band can absorb energy from an electric field and, together with the resulting defect electrons, i.e., holes from the valence band, make the material conductive.

In X-ray detection, the photon emits the entire charge corresponding to its energy at one point in the matter due to a photoelectric effect, which leads to the formation of secondary electrons [32]. A primary electron is lifted from the valence band into the conduction band. Due to its high kinetic energy, numerous secondary electrons and phonons are formed in the process. At higher photon energies (50 keV to 1 MeV), the Compton effect also occurs, in which only part of the energy is transferred to the electron and deposited in the detector.

These generated electron–hole pairs are electrically amplified and can be read out. In the manufacture and structuring of semiconductor detectors, the combination of different conductive areas, doping, is used. Electric fields are applied across these, and the charge carriers generated by the radiation can be transported or amplified within the semiconductor. The principle of operation is that of a diode (Figure 3).

There is a much higher density of conduction electrons in the n-type region of the semiconductor and vice versa in the p-type region. The junction represents a discontinuity in electron density, and a net diffusion from the high-density side to low density occurs for both electrons and holes. The effect of the diffusion from each side of the junction is to build up a net negative space charge on the p-type side and a net positive space charge on the n-type side of the junction, compared to the rest of the p- and n-type. The accumulated space charge creates an electric field that reduces the tendency for further diffusion. At equilibrium, the field is just adequate to prevent additional diffusion across the junction and a steady state charge distribution is established. The region over which the imbalance occurs is called the depletion region [32].

This depletion region acts as a radiation detector. The electric field sweeps electrons created in or near the junction back toward the n-type material, and any holes are swept back toward the p-type material; the motion of these electrons and holes creates the electric signal. However, the contact potential is too low to generate electric fields that will move charge carriers quickly. This can lead to incomplete charge collection, as charges have time to be trapped or recombine. This means that the noise characteristics of an unbiased junction is poor. In addition, the thickness of the depletion region is quite small, so only a small volume of the crystal acts as a radiation detector, which requires an extension of the depletion region [33].

As a general rule, semiconductors with larger atomic masses absorb more X-ray photons, which enhances the response of the detector, but tend to have a smaller band gap energy, which degrades performance in terms of dark current. In addition, low band gap materials are generally more mechanically fragile and prone to damage by irradiation than wide band gap materials [34]. The collection of electrons is better than that of holes, due to their larger mobility and the residual n-type doping in GaN. Some holes become trapped, resulting in a positive charge in the layer. When the bias voltage is large enough to produce a dark current comparable to the photocurrent, the contacts inject additional electrons to balance the positive trapped charge, until electrons recombine with the trapped holes. The longer the electron trapping time, the greater the additional current, the greater the photoconductive gain and the longer the response time. This behaviour is general and common. It must be distinguished from what was observed in [35], where strong nonlinearities and quenching by visible light were observed, and explained by defects activated by X-ray illumination. The GaN was undoped, and the dark currents were low and completely dominated by tunnel currents, which are highly sensitive to defects. This is not the case when the GaN is doped, and the currents are quasi-ideally described by thermionic emission [36]. A low energy is required to generate an electron–hole pair in semiconductor materials (−3 eV for germanium) compared to the energy required to generate an electron-ion pair in gases (−30 eV for typical gas detectors) or to generate an electron–hole pair in scintillators (−100 eV) [32].

As a result, a large number of electron–hole pairs are produced and reach the electrodes, increasing the number of pairs per pulse and reducing both statistical fluctuation and the signal/noise ratio in the intensifier. This is a major advantage over other detectors, and the output pulse provides much better energy resolution. Moreover, the small sensitive area used to detect radiation (a few millimetres) and the high velocity of charge carriers provide an excellent charge collection time (−10^−7^ s).

At this moment, when ionizing radiation interacts with the semiconductor in this depleted region, electrons are lifted into the conduction band, leaving holes in the valence band and creating a large number of electron–hole pairs. When a voltage is applied to the semiconductor, these carriers are readily attracted to the electrodes, and a current flows into the circuit, resulting in a pulse. The size of the pulse is directly proportional to the number of carriers collected, which in turn is proportional to the energy deposited in the material by the incident radiation. In semiconductors, electrons can be also thermally excited from the valence band to the conduction band as the temperature increases. Therefore, some semiconductor detectors must be cooled to reduce the number of electron–hole pairs in the crystal in the absence of radiation. Table 1 summarizes and compares relevant features characterizing a radiation sensor based on the diode principle, the GaN-HEMT and the 2DEG-intensifier.

Another important parameter for qualifying a semiconductor material for X-ray imaging is the signal to noise ratio (S/N)—the higher the S/N ratio, the better. The band gap is a critical factor. On the one hand a large signal is produced by many electron–hole pairs. This can be achieved by low ionizing energies in small band gap materials. On the other hand, only a few intrinsic charge carriers are generated in materials with wide band gap, resulting in low noise.

Before using the GaN-HEMT devices for X-ray imaging, these sensors showed their potential in particle and electromagnetic radiation sensing. A summary review of AlGaN sensing applications can be found in Upadhyay et al. [37]. A comprehensive review was published by Wang et al. [22]. *α*-particle detection was first realized using a GaN double Schottky contact device. Detection was performed using a 5.48 MeV *α*-particles emitted from an ^241^Am source [38]. In addition, 500 µm thick bulk GaN-sensors in a sandwich structure showed *α*-particle detection [39].

When operating the open-gate GaN-HEMT sensor in a mesa structure formation, where an ohmic contact is realized on the cap layer, the device can detect β^−^-radiation and functions as a dosimeter [40]. During exposure of the device to β^−^-particles, energy is transferred in either ionizing or non-ionizing interactions [41]. In ionizing events, (e^+^/e^−^) pairs can be produced within the sensor. The pairs generated in the vicinity of the heterojunction are separated by the internal electrical field, and the electrons subsequently accumulate in the 2DEG, contributing to its charge carrier density. Positive charges, on the other hand, drift into the bulk or towards the sensor surface, in turn contributing to the bulk conductivity. The resulting charge separation causes a photovoltage, that affects the position of the 2DEG relative to the Fermi level, thereby capable of considerably altering the charge carrier density considerably and thus giving rise to high gains [16].

Detection of X-rays with GaN thin film is difficult due to the low absorption coefficient, and both theoretical calculations and experimental measurements show that the absorption coefficient is only large for photon energies between 10 and 20 keV [21]. Our GaN thin film high-mobility heterostructures were tested with X-ray sources from 40 to 300 keV Bremsstrahlung. We found that the photoconductive device response exhibits a large gain, is nearly independent of the angle of radiation, and is constant within 2% of the signal throughout this medical diagnostic X-ray range, indicating that these sensors do not require recalibration for geometry or energy [42].

## 4. GaN-HEMT Based X-ray Imaging

Our GaN-HEMT sensors can operate in two different modes, depending on the X-ray tube voltage. For detection of higher acceleration voltages, tested from 40 to 150 kV, the GaN thin film works as the detector, while for low voltages, tested from 5 to 7 kV, the HEMT structure is employed. A detailed description and characterization of the GaN-HEMT device can be found in Hofstetter et al. [42] and Howgate et al. [16].

The imaging potential of the GaN-HEMT mesa structure was demonstrated using human phantom segments. For this purpose, individual sensors were attached to a two-dimensional translation stage (SH4018L1704 stepping motors, Telco, Houston, TX, USA) and scanned under human phantom segments, while simultaneously recording the device response. An acceleration voltage of 150 kV was used. Positioning of the translation stage, source meter, and data acquisition were controlled by a LabView program 8.6 code (National Instruments, Austin, TX, USA). The source meter provides the possibility of simultaneously biasing the detector (U_SD_) and measuring the current (I_SD_). The integration time for a single data reading is characterized by the number of power line cycles, NPLC. Here, one PLC at 50 Hz results in 20 ms, and integration time was set to NPLC one. Measurements were performed with a data acquisition frequency in the 10 Hz regime. Unlike other semiconductor detectors, images were recorded at room temperature without an additional cooling of the sensor material. Although such devices have the potential to be easily miniaturized to form high-density pixel arrays for imaging purposes, the chips we fabricated were 5 × 3 mm^2^, with an internal active area of 2.8 × 1 mm^2^, and only contained one single device, thus limiting the pixel dimensions to this size. Several hundred pictures were acquired at different offsets from the centre position of each pixel, with an acquisition time of 200 ms, and the recorded data were recombined into a single image by interpolation and averaging. The two-dimensional data were plotted on an 8-bit X-ray scale contour map (Figure 4) [23,43]. A detailed description of the used mesa HEMT sensors and experimental results on X-ray dosimetry and imaging are presented in Hofstetter et al. [42].

Unlike most established semiconductor X-ray detectors, in which photo-excited electron–hole pairs are separated and collected via an internal space charge region, our devices are operated without the intentional formation of a depletion layer. The photocurrents of GaN-sensors are monitored with the application of a small DC voltage between two ohmic contacts. Although space charge regions are present, due to surface band bending, interface band alignment, and extended defects, the measured current primarily flows in parallel to such depletion regions, rather than through them. Illumination leads to non-equilibrium concentrations of free carriers, which reduce the total volume of the depletion regions, and, in turn, increase the effective volume of the material through which carriers can be transported, i.e., both conductivity and conductance are increased. As a result, the photocurrent response is not directly limited by the direct conversion of absorbed energy into free charge carriers, and the devices can function as X-ray sensitive sensors with significant internal intensification. Due to the light and temperature sensitivities of the devices, all measurements were performed with careful exclusion of stray light and at room temperature [42].

Compared to other semiconductor detectors, which are based on a measurable current due to the separation of electron–hole pairs, our GaN sensors show effects that can be described with this simple model of charge carrier generation, as in photovoltaics.

The GaN-HEMT works as a direct X-ray converter, and the mode of operation is based on the photo effect (photoconductive or photoelectric effect). The photo effect summarizes three closely related but different processes of photon interaction with matter. In all three cases, an electron is released from a bond, i.e., in an atom, in the valence band, or in the conduction band of a solid material, dissolved by absorbing a photon. The energy of the photon must be at least as high as the binding energy of the electron (photo- or Compton effect).

A distinction is made between the external and internal photoelectric effect (Auger effect). The latter is crucial for the GaN-HEMT. In the case of the internal photo effect, a bound shell electron is released in a solid through photon absorption, but without leaving the body. This occurs in semiconductor materials and insulators. The photoelectrons and the holes created at the same time increase the electrical conductivity.

Photoconductivity is the increase in the electrical conductivity of semiconductor materials due to the formation of unbound electron–hole pairs during irradiation. The electrons are lifted from the valence band into the energetically higher conduction band by means of the energy of the photons, for which the energy of the individual photon must at least correspond at least to the band gap of the irradiated semiconductor. Since the size of the band gap depends on the nature of the material, the maximum wavelength of the light up to which photoconduction occurs differs depending on the semiconductor.

## 5. GaN/AlGaN HEMT—2DEG Intrinsic Intensifier

The GaN-HEMT sensor follows the basic principle of a metal-oxide-semiconductor field effect transistor (MOS-FET), a voltage-controlled circuit element whose conductance is controlled via the gate-source voltage. The intrinsic intensifier, such as a HEMT consisting of a heterostructure of GaN (E_g_ = 3.4 eV) and Al_0.26_Ga_0.74_N (E_g_ = 4 eV), operates accordingly. Since the band gap of AlGaN is larger than that of GaN, a 2DEG is formed at the interface of these two materials on the GaN side, which can serve as a conductive channel. One of the most interesting features of Al_x_Ga_1-x_N is the possibility of tailoring the energy gap, and thus the inherent 2DEG, by varying the Al concentration, with Al working as a dopant. When using the energy band model, the piezoelectric polarization and the resulting internal electric fields occurring in the AlGaN/GaN interface cause the conduction band to bend below the Fermi level, resulting in the accumulation of free electrons in a potential well (Figure 5). Typically, the 2DEG is characterized by sheet carrier density values in the order of 10^13^ cm^−2^ and mobility in the range of 1000-2000 cm^2^/Vs [10].

With regard to the operating principle of AlGaN/GaN heterostructure, X-rays change the conductivity of the 2DEG. When an electron–hole pair is generated, this results in charge separation due to the built-in electric field, perpendicular to the heterojunction plane. As a result, electrons drift into the 2DEG channel, and the holes drift into the volume and surface regions of the sensor. Since the GaN-HEMT sensors are more or less transparent, the absorbed energy is roughly constant as a function of depth, comparable to the linear energy transfer, through the sensor structure, and most of the electron–hole pairs are generated inside the much thicker GaN-buffer layer.

This charge separation, and the associated accumulation of electrons in the 2DEG channel, leads to an internal photovoltaic effect that causes a shift in the threshold potential. As a result, the current response is proportional to the transconductance of the detector. At a constant applied voltage, the current through the 2DEG forms a signal that can be assigned to a dose rate. The layer structure is only a few nm thick, and the 2DEG has no three-dimensional extension. Charge carriers generated by photons during X-ray imaging in the GaN-buffer layer can now additionally fall into this potential well and form an inversion layer with high carrier density and mobility. This has a direct impact on the photo-generated current. 

Thus, an area-dependent, or volume-independent, measurement takes place in the HEMTs, which is a decisive criterion, especially for high-resolution dosimetry (e.g., the formation of artifacts, or distortion of the measurement signal). The measurement signal in the 2DEG is generated in <10^−3^ s. Due to the specific band structure and the piezoelectric nature of both semiconductor materials, a triangular potential well is generated at the interface. An important boundary condition is the constant Fermi level over the entire structure (dotted red line, Figure 5).

Since the 2DEG is naturally present in the AlGaN/GaN heterostructure, and the Fermi level at the interface lies above the conduction band minimum (Figure 5), a current flows between source and drain, even when the gate bias is zero. The output current of a HEMT can be modulated by applying a negative bias voltage to the gate, until a “threshold voltage” is reached, at which the Fermi level is pulled below the conduction band edge of the AlGaN layer and the 2DEG channel is depleted (Figure 5).

In summary, the triangular potential well is our tuneable switch for sensitivity; the intrinsic intensifier (Figure 6). This means that the collecting of electrons and the constant Fermi level can be changed by gate voltage [44].

## 6. GaN/AlGaN HEMT Based X-ray Imaging

To demonstrate the potential of the intrinsic intensification principle of the GaN- HEMT sensor, a single GaN/AlGaN HEMT sensor was connected to a computer-controlled two-dimensional translation stage, to image the light bulb of a small flashlight. Scans were performed using a Pt-gated device with a 500 µm HEMT channel. Source–drain bias was set to 120 mV, and the gate was biased to V_G_ = 3.5 V for intrinsic intensification. Control and data acquisition were the same as described in Section 4. As indicated in Figure 7, the spiral filament of the light bulb is clearly visible, and in the surrounding area the increase of current shows an absence of material, indicating that the centre is hollowed out to form a sealed cavity. In this proof-of-concept, we were able to achieve a detection range down to 50 µm, with a photon rate in the 10^6^ counts per second regime. The acquisition time for a single pixel was about 1 s due to motor movement and data acquisition time.

## 7. Summary and Conclusions

The fabrication of GaN heterostructures with improved quality, at economic scale, is now mature. GaN-based devices are capable of significantly outperforming their silicon-based counterparts. To date, little attention has been paid to GaN-based X-ray detection. However, we believe that the introduction of a 2DEG channel, by adding AlGaN/GaN heterointerfaces into the GaN thin film, will open another chapter in *GaNification* [10], as the tremendous sensitivity of GaN HEMT sensors will reach a new level of hardware innovation, reducing the effective dose used today in routine X-ray and CT scans. Device dimensions, i.e., pixel size, can be reduced by orders of magnitude using modern standard microfabrication techniques. Thus, there should be no practical constraints on the production of large-area X-ray sensors based on pixel arrays. High-resolution imaging using the shown integrated GaN HEMT pixel sensors should be possible based on our proof-of-principle measurements [44,45]. The material properties of GaN heterostructures such as robustness, light weight, and features such as room temperature operation without cooling, excellent signal-to-noise ratio, and the 2DEG-based tuneable intrinsic sensitivity, altogether form the basis for enabling a paradigm shift in X-ray detector technology.

## Figures and Tables

**Figure 1 micromachines-13-00147-f001:**
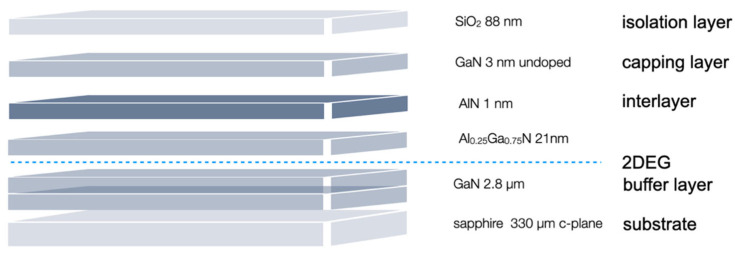
Schematic drawing of the GaN HEMT heterostructure. Dotted line shows the 2DEG area.

**Figure 2 micromachines-13-00147-f002:**
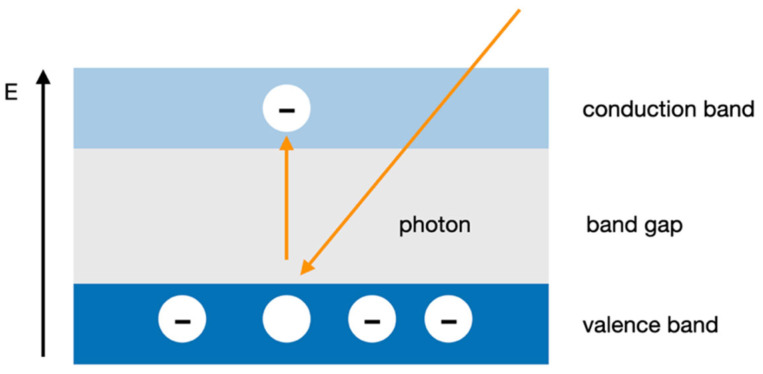
Principle of the formation of electron–hole pairs with ionizing radiation. Bands determine the density of available energy states. E = energy of electrons.

**Figure 3 micromachines-13-00147-f003:**
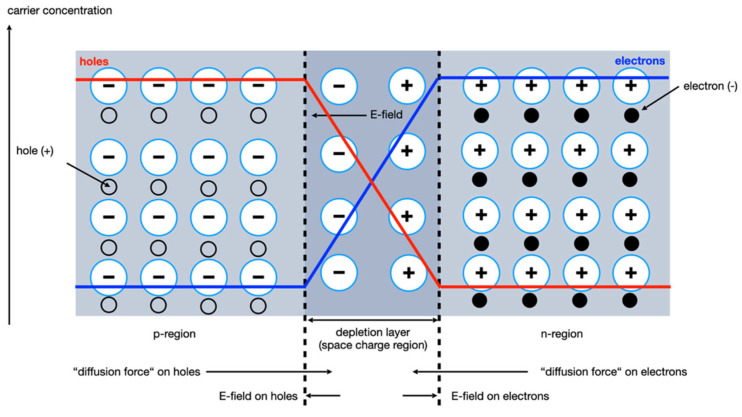
Schematic sketch of a p-n junction.

**Figure 4 micromachines-13-00147-f004:**
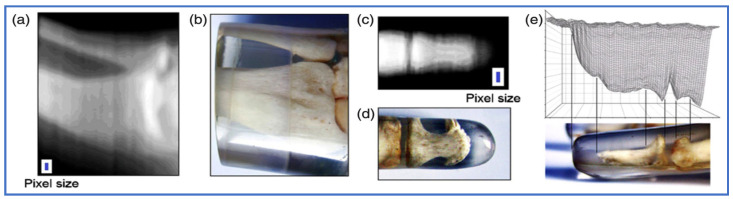
The 8-bit X-ray grey-scale contour maps recorded using GaN-HEMTs (the total device size is indicated (white) and the active area (blue)) of a human (**a**) wrist and (**c**) index finger phantom. Optical pictures of the scanned (**b**) wrist and (**d**) finger phantom. (**e**) Side view of a 3D plot of the data in (**c**) compared to an optical image in the same orientation ([42]; copyright IOP Publishing, 2011).

**Figure 5 micromachines-13-00147-f005:**
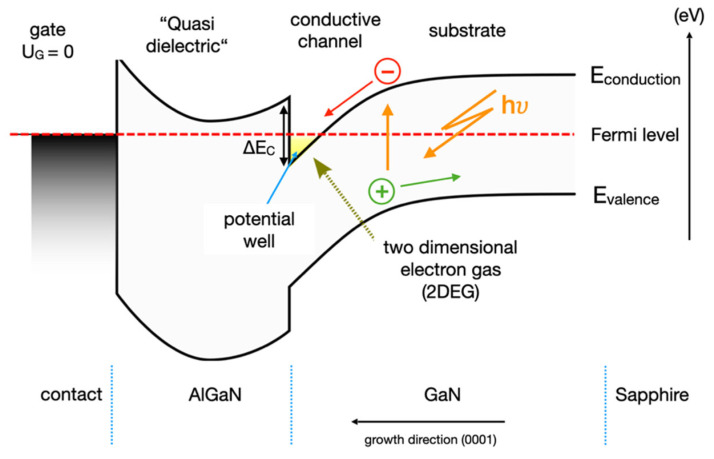
Energy band diagram of a HEMT with 2D electron gas. Schematic representation of the band-bending of the conductive band at the GaN/AlGaN material interface. The conduction band is bent below the Fermi level, which leads to an accumulation of free electrons in the potential well.

**Figure 6 micromachines-13-00147-f006:**
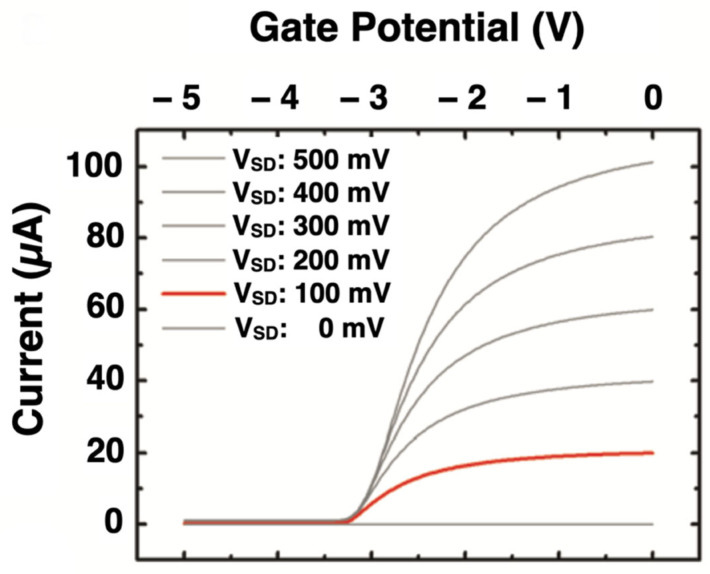
A plot of the source–drain current, as a function of gate–drain potential (reproduced with permission from [16]; published by Wiley Online Library, 2012).

**Figure 7 micromachines-13-00147-f007:**
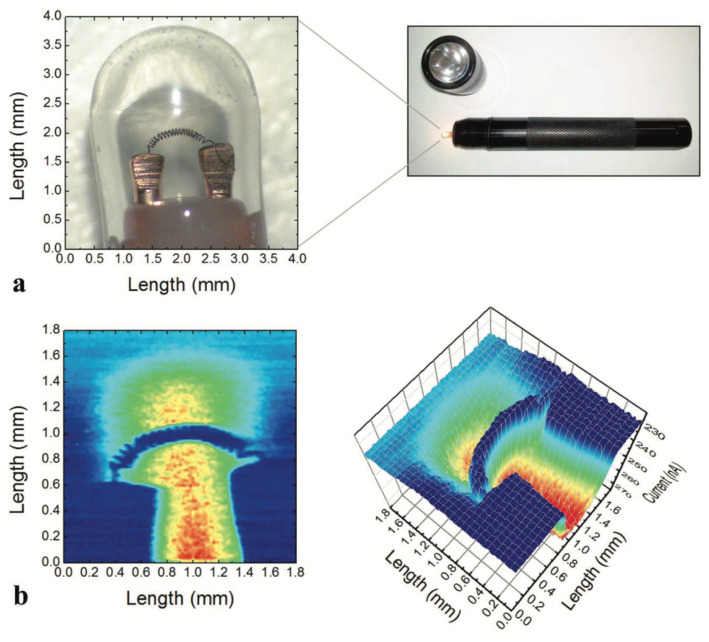
(**a**) A microscope image of the bulb shows the approximately 3 × 3 mm object with a 50 μm thick spiral filament. (**b**) High resolution HEMT X-ray image of a flashlight bulb. The 2D and 3D images show the plotted X-ray data. Scans were performed with a Pt-gated device with a 500 µm HEMT channel. The source–drain bias was set to 120 mV and the gate was biased to V_G_ = −3.5 V. All measurements were performed with 5 keV X-rays at room temperature. Reproduced with permission from [43].

**Table 1 micromachines-13-00147-t001:** Comparison of the functional principle of the diode, GaN-HEMT, and 2DEG intensifier.

Feature	Diode	GaN-HEMT	2DEG Intensifier
Contacts	Schottky ohmic, with p-n, pin-diodes	ohmic	ohmic
Detection principle	diode	FET	MOSFET
Depletion layer	yes	no	no
Current flow	no	yes	yes
Aging	yes	no	no
Quality feature	dark current	signal/noise ratio	signal/noise ratio
Increasing sensitivity	enlarge depletion layer	adaption of electrical conductivity	alteration of gate voltage

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
