# Peer review of "GaN Heterostructures as Innovative X-ray Imaging Sensors—Change of Paradigm"

_micromachines, 2022, doi:10.3390/mi13020147_

Round 1
Reviewer 1 Report
This paper reviewed existing technology x-ray sensors based on GaN film or GaN HEMT. It is an interesting work, but a major revision is required before published.
1) Sensor structures, or schematics are needed for the reader to understand the difference of difference techniques.
2) The signal interpretation from electrical signal to optical signal shall be given. For example, how to extract optical information from Fig.5. For instance by Vt shift or by Ron. It is not clear.
3) There is room to shorten the length of the paper. For example from line 315 to 333. The author explained how the 2DEG is formed by rephrasing 3 times without added value.
Author Response
Thank you very much for the reviewer’s comments. We appreciate their comments and here are our in the text implemented corrections. You can follow these by track following.
This paper reviewed existing technology x-ray sensors based on GaN film or GaN HEMT. It is an interesting work, but a major revision is required before published.
1) Sensor structures, or schematics are needed for the reader to understand the difference of difference techniques.
We added a schematic drawing (figure 1) of our GaN heterostructure and explained the processing of the waver in more detail (line 86-99).
2) The signal interpretation from electrical signal to optical signal shall be given. For example, how to extract optical information from Fig.5. For instance by Vt shift or by Ron. It is not clear.
We added in our manuscript a more detailed description of the data acquisition (line 340 to 359).
3) There is room to shorten the length of the paper. For example from line 315 to 333. The author explained how the 2DEG is formed by rephrasing 3 times without added value.
According to reviewer’s comment we shortened chapter 5.
We hope our changes correspond the reviewer’s comments and our manuscript is ready for publication in Micromachines.
Kind regards,
PD Dr. Stefan Thalhammer

Reviewer 2 Report
The work of Thalhammer et al. reports a review on GaN Heterostructures as Innovative X-ray Imaging Sensors. This manuscript requires major revisions and clarifications. The authors need to address the listed concerns and comments to improve its quality.
- Though it is a review manuscript, the number of references is not sufficient. Thus, the Reviewer encourages to add more relevant and recent works on GaN HEMT in the revised manuscript.
- In addition to the Transfer characteristic plot in Figure 5, please also add/ refer to the other types of response plots i.e., output characteristics, etc.
- 6 and the relevant explanation in the manuscript are completely obscure. Please rewrite/clarify the entire section 6. For example, authors have claimed that “High-resolution HEMT x-ray image of a flashlight bulb” what do authors mean by “High-resolution HEMT x-ray image”, while the scale bar is in mm. Similarly, Figure. 6a is an optical image.
Please explain the following concerns in section 6 in detail:
- two-dimensional translation stage for imaging. What type of stage, how it was operated piezo-driven, servo motor driven or any other mechanism. How it is related to the imaging system.
- “increase of current shows an absence of material indicating” please explain this sentence
- How images were taken, please explain in detail
- How the output current is related to the Figure 6b, how it was converted to image/bitmap.
Author Response
Thank you very much for the reviewer’s comments. We appreciate their comments and here are our in the text implemented corrections. You can follow these by track following.
The work of Thalhammer et al. reports a review on GaN Heterostructures as Innovative X-ray Imaging Sensors. This manuscript requires major revisions and clarifications. The authors need to address the listed concerns and comments to improve its quality.
1. Though it is a review manuscript, the number of references is not sufficient. Thus, the Reviewer encourages to add more relevant and recent works on GaN HEMT in the revised manuscript.
We included more references in the fabrication chapter. In our opinion we cited all relevant publications. Due to the novelty of our topic, namely the role of GaN heterostructures in x-ray sensing only a few papers are published to date.
2. In addition to the Transfer characteristic plot in Figure 5, please also add/ refer to the other types of response plots i.e., output characteristics, etc.
Former figure 5, now current figure 6, shows a plot of the source–drain current as a function of gate–drain potential. This demonstrates the operation mode of the GaN heterostructurestructure and the possibility to tune the sensitivity of our intrinsic intensifier of the GaN heterostructure. While recording this plot no ionizing radiation was involved.
3. 6 and the relevant explanation in the manuscript are completely obscure. Please rewrite/clarify the entire section 6. For example, authors have claimed that “High-resolution HEMT x-ray image of a flashlight bulb” what do authors mean by “High-resolution HEMT x-ray image”, while the scale bar is in mm. Similarly, Figure. 6a is an optical image.
We clarified and updated said chapter. The figure caption was corrected according the reviewer’s request. We also added a more precise description of the data acquisition and processing as well as details of the 2-dimensional stage.
In addition, we have made some slight changes with respect to the wording of section 7. To that end we have included some literature citations and further emphasized the superiority of the GaN heterostructure for x-ray detector technology.
We hope our changes correspond the reviewer’s comments and our manuscript is ready for publication in Micromachines.
Kind regards,
PD Dr. Stefan Thalhammer

Round 2
Reviewer 2 Report
I have reviewed the manuscript; it can be accepted.